# REVISITING ADAPTERS WITH ADVERSARIAL TRAINING

**Sylvestre-Alvise Rebuffi, Francesco Croce**[*] **& Sven Gowal**
DeepMind, London
`{sylvestre,sgowal}@deepmind.com`

## ABSTRACT

While adversarial training is generally used as a defense mechanism, recent works show that it can also act as a regularizer. By co-training a deep network on clean and adversarial inputs, it is possible to improve classification accuracy on the clean, non-adversarial inputs. We demonstrate that, contrary to previous findings, it is not necessary to separate batch statistics when co-training on clean and adversarial inputs, and that it is sufficient to use adapters with few domain-specific parameters for each type of input. We establish that using the classification token of a Vision Transformer (VIT) as an adapter is enough to match the classification performance of dual normalization layers, while using significantly less additional parameters. First, we improve upon the top-1 accuracy of a non-adversarially trained VIT-B16 model by +1.12% on IMAGENET (reaching 83.76% top-1 accuracy). Second, and more importantly, we show that training with adapters enables *model soups* through linear combinations of the *clean* and *adversarial* tokens. These *model soups*, which we call *adversarial model soups*, allow us to trade-off between clean and robust accuracy without sacrificing efficiency. Finally, we show that we can easily adapt the resulting models in the face of distribution shifts. Our VIT-B16 obtains top-1 accuracies on IMAGENET variants that are on average +4.00% better than those obtained with Masked Autoencoders.

## 1 INTRODUCTION

Deep networks are inherently susceptible to adversarial perturbations. Adversarial perturbations fool deep networks by adding an imperceptible amount of noise which leads to an incorrect prediction with high confidence (Carlini & Wagner, 2017; Goodfellow et al., 2015; Kurakin et al., 2016b; Szegedy et al., 2014). There has been a lot of work on building defenses against adversarial perturbations (Papernot et al., 2016; Kannan et al., 2018); the most commonly used defense is adversarial training as proposed by Madry et al. (2018) and its variants (Zhang et al., 2019; Pang et al., 2020; Huang et al., 2020; Rice et al., 2020; Gowal et al., 2020), which use adversarially perturbed images at each training step as training data. Earlier studies (Kurakin et al., 2016a; Xie et al., 2019b) showed that using adversarial samples during training leads to performance degradation on clean images. However, AdvProp (Xie et al., 2019a) challenged this observation by showing that adversarial training can act as a regularizer, and therefore improve nominal accuracy, when using dual batch normalization (BatchNorm) layers (Ioffe & Szegedy, 2015) to disentangle the clean and adversarial distributions.

We draw attention to the broad similarity between the AdvProp approach and the adapters literature (Rebuffi et al., 2017; Houlsby et al., 2019) where a single backbone network is trained on multiple domains by means of adapters, where a few parameters specific to each domain are trained separately while the rest of the parameters are shared. In light of this comparison, we further develop the line of work introduced by AdvProp and analyze it from an adapter perspective. In particular, we explore various adapters and aim to obtain the best classification performance with minimal additional parameters. Our contributions are as follows:

- We show that, in order to benefit from co-training on clean and adversarial samples, it is not necessary to separate the batch statistics of clean and adversarial images in BatchNorm layers. We demonstrate empirically that it is enough to use domain specific trainable parameters to achieve similar results.

---

[*]Work done during an internship at DeepMind

- Inspired by the adapters literature, we evaluate various adapters. We show that training separate classification tokens of a VIT for the clean and adversarial domains is enough to match the classification performance of dual normalization layers with $49\times$ fewer domain specific parameters. This classification token acts as a conditioning token which can modify the behaviour of the network to be either in *clean* or *robust mode* (Figure 1).

- Unlike Xie et al. (2019a) and Herrmann et al. (2022), we also aim at preserving the robust performance of the network against adversarial attacks. We show that our conditional token can obtain SOTA nominal accuracy in the *clean mode* while at the same time achieving competitive $\ell_\infty$-robustness in the *robust mode*. As a by-product of our study, we show that adversarial training of VIT-B16 on IMAGENET leads to state-of-the-art robustness against $\ell_\infty$-norm bounded perturbations of size $4/255$.

- We empirically demonstrate that training with adapters enables *model soups* (Wortsman et al., 2022). This allow us to introduce *adversarial model soups*, models that trade-off between clean and robust accuracy through linear interpolation of the clean and adversarial adapters. To the best of our knowledge, our work is the first to study *adversarial model soups*. We also show that *adversarial model soups* perform better on IMAGENET variants than the state-of-the-art with masked auto-encoding (He et al., 2022).

## 2 RELATED WORK

**Adversarial training.** Although more recent approaches have been proposed, the most successful method to reduce the vulnerability of image classifiers to adversarial attacks is *adversarial training*, which generates on-the-fly adversarial counterparts for the training images and uses them to augment the training set (Croce et al., 2020). Goodfellow et al. (2015) used the single-step Fast Gradient Sign Method (FGSM) attack to craft such adversarial images. Later, Madry et al. (2018) found that using iterative Projected Gradient Descent (PGD) yields models robust to stronger attacks. Their scheme has been subsequently improved by several modifications, e.g. a different loss function (Zhang et al., 2019), unlabelled or synthetic data (Carmon et al., 2019; Uesato et al., 2019; Gowal et al., 2021), model weight averaging (Gowal et al., 2020), adversarial weight perturbations (Wu et al., 2020), and better data augmentation (Rebuffi et al., 2021). While the main drawback of adversarial training is the degradation of performance of robust models on clean images (Tsipras et al., 2018), Xie et al. (2019a) showed that adversarial images can be leveraged as a strong regularizer to *improve* the clean accuracy of classifiers on IMAGENET. In particular, they propose AdvProp, which introduces separate BatchNorm layers specific to clean or adversarial inputs, with the remaining layers being shared. This approach and the role of normalization layers when training with both clean and adversarial points has been further studied by (Xie & Yuille, 2019; Walter et al., 2022). Recently, Wang et al. (2022) suggest removing BatchNorm layers from the standard RESNET architecture (He et al., 2016) to retain high clean accuracy with adversarial training, but this negatively affects the robustness against stronger attacks.[1] Finally, (Kireev et al., 2021; Herrmann et al., 2022) showed that carefully tuning the threat model in adversarial training might improve the performance on clean images and in the presence of distribution shifts, such as *common corruptions* (Hendrycks & Dietterich, 2018).

**Adapters.** In early work on deep networks, Caruana (1997) showed that sharing network parameters among tasks acts as a regularizer. Aiming at a more efficient parameter sharing, (Rebuffi et al., 2017; Rosenfeld & Tsotsos, 2018) introduced adapters – small training modules specific to each task which can be stitched all along the network. In other lines of work, (Mallya et al., 2018; Mancini et al., 2018) adapt a model to new tasks using efficient weight masking and (Li et al., 2016; Maria Carlucci et al., 2017) perform domain adaptation by batch statistics modulation. While these approaches require having as many adapters as tasks, Perez et al. (2018) propose an adapter layer whose weights are generated by a conditioning network. Besides computer vision, adapters are also used in natural language processing for efficient fine-tuning (Houlsby et al., 2019; Pfeiffer et al., 2020; Wang et al., 2020) and multi-task learning (Stickland & Murray, 2019).

**Merging multiple models.** While ensembles are a popular and successful way to combine multiple independently trained classifiers to improve on individual performance (Ovadia et al., 2019; Gontijo-Lopes et al., 2021), they increase the inference cost as they require a forward pass for each sub-network

---

[1] See https://github.com/amazon-research/normalizer-free-robust-training/issues/2.

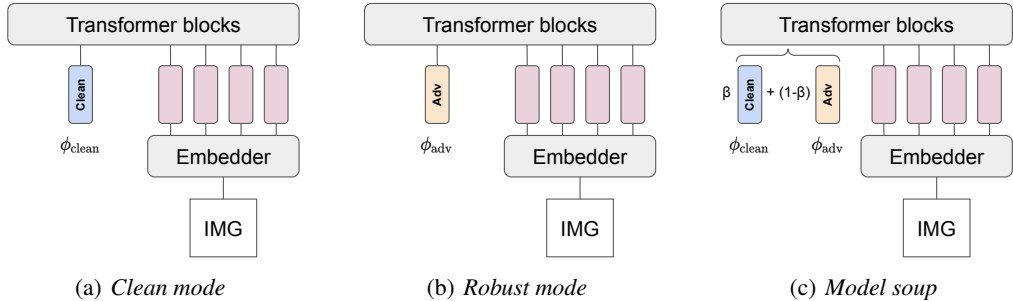

(a) *Clean mode*  (b) *Robust mode*  (c) *Model soup*

**Figure 1: Classification token as adapter.** The image is embedded into visual tokens (in pink). The behaviour of the model can set to the *clean mode*, *robust mode* or a *model soup* by respectively using the *clean* token (in blue), the *adversarial* token (in orange) or a linear combination of these two tokens. The parameters of the embedder, the transformer blocks and the classifier are shared between modes.

of the ensemble. An alternative approach is taken by Wortsman et al. (2022) who propose to fine-tune a fully trained model with different hyperparameter configurations and then average the entire set of weights of the various networks. The obtained *model soups* get better performance than each individual model and even ensembles. Model soups are in spirit similar to Stochastic Weight Averaging (Izmailov et al., 2018) which consists in averaging weights along an optimization trajectory rather than averaging over independent runs.

## 3  METHOD

### 3.1  CO-TRAINING WITH NOMINAL AND ADVERSARIAL TRAINING

Goodfellow et al. (2015) propose adversarial training as a way to regularize standard training. They jointly optimize the model parameters $\boldsymbol{\theta}$ on clean and adversarial images with the co-training loss

$$\alpha L(f(\boldsymbol{x};\boldsymbol{\theta}),y) + (1-\alpha)\max_{\boldsymbol{\delta}\in\mathbb{S}} L(f(\boldsymbol{x}+\boldsymbol{\delta};\boldsymbol{\theta}),y), \tag{1}$$

where pairs of associated examples $\boldsymbol{x}$ and labels $y$ are sampled from the training dataset, $f(\cdot;\boldsymbol{\theta})$ is a model parametrized by $\boldsymbol{\theta}$, $L$ defines the loss function (such as the cross-entropy loss in the classification context), and $\mathbb{S}$ is the set of allowed perturbations. Setting $\alpha = 1$ boils down to nominal training on clean images and setting $\alpha = 0$ leads to adversarial training as defined by Madry et al. (2018). In our case, we consider $\ell_\infty$ norm-bounded perturbations of size $\epsilon = 4/255$, so we have $\mathbb{S} = \{\boldsymbol{\delta} \mid \|\boldsymbol{\delta}\|_\infty \leq \epsilon\}$, and we use untargeted attacks to generate the adversarial perturbations $\boldsymbol{\delta}$ (see details in Section 4).

### 3.2  SEPARATING BATCH STATISTICS IS NOT NECESSARY

BatchNorm is a widely used normalization layer shown to improve performance and training stability of image classifiers (Ioffe & Szegedy, 2015). We recall that a BatchNorm layer, given a batch as input, first normalizes it by subtracting the mean and dividing by the standard deviation computed over the entire batch, then it applies an affine transformation, with learnable scale and offset parameters. During training, it accumulates these so-called *batch statistics* to use during test time, so that the output of the classifier for each image is independent of the other images in the batch. The batch statistics can be seen an approximation of the statistics over the image distribution.

Xie et al. (2019a) show that optimizing the co-training loss in Eq. 1 can yield worse results on clean images than simple nominal training. This is especially the case when the network has a low capacity or the attack (i.e., the inner maximization) is too strong (such as using a large perturbation radius $\epsilon$). To solve this issue, they propose AdvProp, which consists in using distinct BatchNorm layers for clean and adversarial images. They argue that *"maintaining one set of [BatchNorm] statistics results in incorrect statistics estimation"*, which could be the reason for the performance degradation. We note that using two sets of BatchNorm layers for the clean and adversarial samples as in AdvProp

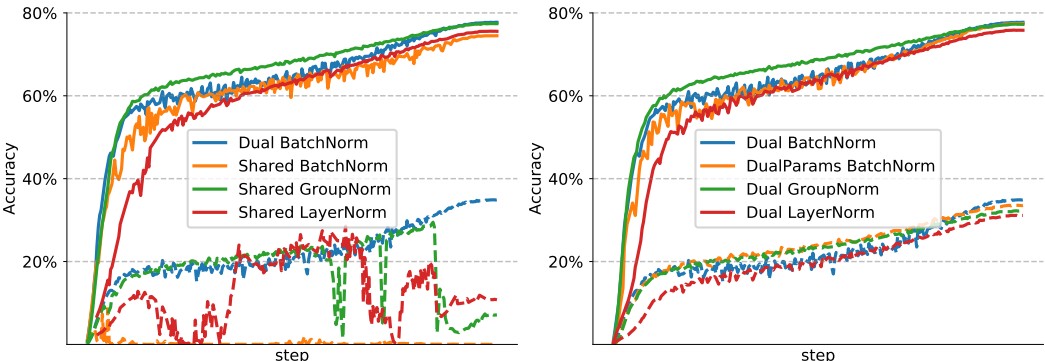

**Figure 2: Dual parameters are enough.** We report the clean (solid lines) and robust accuracy (dashed lines) over training steps of RESNET-50 trained on IMAGENET with the co-training loss of Eq. 1 ($\epsilon = 4/255$): for models with dual layers. Clean accuracy refers to the *clean mode* and the robust accuracy to the *robust mode*. *Left panel.* We compare models with different normalization layers with no domain-specific parameters (Shared BatchNorm, Shared LayerNorm, Shared GroupNorm) to Dual BatchNorm as proposed by Xie et al. (2019a): regardless of the type of normalization, the robustness of classifiers without dual layers drops to (almost) zero at the end of training. *Right panel.* We use domain-specific normalization layers (Dual BatchNorm, Dual LayerNorm, Dual GroupNorm) and a model with BatchNorm with shared batch statistics but domain-specific scale and offset (DualParams BatchNorm): all models achieve high clean and robust accuracy.

creates two sets of batch statistics but also two sets of learnable scale and offset parameters. In the following we investigate whether having separate batch statistics is a necessary condition for successful co-training.

Figure 2 shows the clean and robust accuracy of various model architectures as training progresses. The left panel demonstrates that, if we share both batch statistics and scales/offsets (Shared Batch-Norm, orange curves), the robust accuracy (orange dashed line) quickly drops, far from the one obtained by AdvProp (Dual BatchNorm, blue curve) which is above $34\%$. However, if we use a single set of batch statistics but specific scales and offsets for clean and adversarial images, we can observe on the right panel of Figure 2 that the robust accuracy (DualParams BatchNorm, orange dashed line) matches the one (blue dashed line) obtained by AdvProp. This demonstrates that it is possible to achieve nominal and robust classification results similar to those of AdvProp without separate batch statistics.

Furthermore, there exist normalization layers such as LayerNorm (Ba et al., 2016) or GroupNorm (Wu & He, 2018) which do not use batch statistics, as their normalization step is done per sample and not per batch. Hence, according to the hypothesis of Xie et al. (2019a), these types of normalization layer should not suffer from performance degradation. Nevertheless, the left panel of Figure 2 shows that their robust accuracy (green and red dashed lines) does not match the robust accuracy of AdvProp (Dual BatchNorm), and is unstable over training steps. However, by making the scales and offsets of LayerNorm and GroupNorm specific to clean and adversarial images, their robust accuracy matches that obtained with dual BatchNorm layers, as shown in the right panel of Figure 2. This suggests that a key element to make the co-training loss of Eq. 1 work for various normalization layers is to have trainable parameters which are specific to the clean and adversarial images.[2]

### 3.3 REVISITING ADAPTERS WITH ADVERSARIAL TRAINING

The last observation strongly relates this setting to the adapters literature where a single backbone architecture has some parameters, called adapters, which are specific to different domains while the rest of the parameters are shared among tasks. In our case, the clean images form one domain and the adversarial images constitute another domain. In this work, we go beyond having separate normalization layers for the clean and adversarial images and investigate other types of adapters.

---

[2]Interestingly, contrary to our observation that standard GroupNorm fails to retain robustness, Xie & Yuille (2019) report that GroupNorm matches Dual BatchNorm. We explain this difference as we use a stronger untargeted attack in this manuscript compared to the targeted attack of Xie & Yuille (2019). Using a stronger attack allows us to reveal failure modes that would have been hidden otherwise.

Formally, the model parameters $\boldsymbol{\theta}$ can be decomposed into parameters $\boldsymbol{\psi}$ which are shared among domains and parameters $\boldsymbol{\phi}$ which are specific to a domain. We call $\boldsymbol{\phi}_{\text{clean}}$ the parameters used when training on clean images and $\boldsymbol{\phi}_{\text{adv}}$ the parameters used when training on adversarial images. For example, in Section 3.2, when we used dual LayerNorm layers, the scales and offsets of these normalization layers are contained in $\boldsymbol{\phi}_{\text{clean}}$ and $\boldsymbol{\phi}_{\text{adv}}$ whereas the rest of the model parameters are in $\boldsymbol{\psi}$. Based on Eq. 1, we optimize the following loss:

$$\alpha L(f(\boldsymbol{x}; \boldsymbol{\psi} \cup \boldsymbol{\phi}_{\text{clean}}), y) + (1 - \alpha) \max_{\boldsymbol{\delta} \in \mathbb{S}} L(f(\boldsymbol{x} + \boldsymbol{\delta}; \boldsymbol{\psi} \cup \boldsymbol{\phi}_{\text{adv}}), y). \tag{2}$$

Finally, we introduce some notation for models with adapters at inference time: we call $f(\cdot; \boldsymbol{\psi} \cup \boldsymbol{\phi}_{\text{clean}})$ the *clean mode* for prediction as we use the adapters $\boldsymbol{\phi}_{\text{clean}}$ trained on the clean data. Conversely, we call $f(\cdot; \boldsymbol{\psi} \cup \boldsymbol{\phi}_{\text{adv}})$ the *robust mode* when using the adapters $\boldsymbol{\phi}_{\text{adv}}$ trained on the perturbed data.

### 3.4 TRAINING WITH ADAPTERS ENABLES ADVERSARIAL MODEL SOUPS

Wortsman et al. (2022) propose *model soups*, which consist in averaging the weights of multiple models fine-tuned from the same pre-trained model. The resulting weight averaged model can benefit from the original models without incurring any extra compute and memory cost at inference time. Currently, in our setting the user would have to know at test time if the network should be in *clean* or *robust mode*. A *model soup*, by its ability to merge models, is a way to bypass this issue. We formulate the hypothesis that training with adapters enables *model soups*. With this in mind, we observe that training with adapters means that most of the model parameters are already shared, so *model souping* would simply consist in linearly interpolating the weights of the adapters for the two modes. We call *adversarial model soups*, the model soups with a model co-trained on clean and adversarial samples. We get the following parametrized model:

$$f(\cdot; \boldsymbol{\psi} \cup (\beta \boldsymbol{\phi}_{\text{clean}} + (1 - \beta) \boldsymbol{\phi}_{\text{adv}})) \tag{3}$$

where $\beta$ is the weighting factor when averaging the adapters. If $\beta = 1$, the *model soup* boils down to the *clean mode* and conversely $\beta = 0$ corresponds to the *robust mode*. In Section 5.2, we assess this hypothesis and show that forming *model soups* between independent nominal and robust models fails.

## 4 EXPERIMENTAL SETUP

**Architecture.** We focus our study on the B16 variant of the Vision Transformer (VIT-B16) introduced by Dosovitskiy et al. (2020). We adopt the modifications proposed by He et al. (2022): the linear classifier is applied on the mean of the final tokens except the classification token. We train this network by using supervised training from scratch as proposed in He et al. (2022) (see the appendix).

**Attacks.** We consider adversarial robustness against untargeted $\ell_\infty$-bounded attacks with radius $\epsilon = 4/255$. This is the most common setup for IMAGENET models, and it is more challenging to defend against than the targeted threat model used by Xie & Yuille (2019). To generate the adversarial perturbations we use Projected Gradient Descent (Madry et al., 2018) with 2 steps named $\text{PGD}^2$ (see details in the appendix) at training time and with 40 steps for evaluation ($\text{PGD}^{40}$).

**Datasets.** We focus our experimental evaluation on the IMAGENET dataset (Russakovsky et al., 2015), with images at $224 \times 224$ resolution for both training and testing, as this is the standard large-scale benchmark for SOTA models and was used by Xie et al. (2019a) for AdvProp. We report clean and adversarial accuracy on the whole validation set. Moreover, we test the robustness against distribution shifts via several IMAGENET variants: IMAGENET-C (Hendrycks & Dietterich, 2018), IMAGENET-A (Hendrycks et al., 2019), IMAGENET-R (Hendrycks et al., 2020), IMAGENET-SKETCH (Wang et al., 2019), and Conflict Stimuli (Geirhos et al., 2018).

## 5 EXPERIMENTAL RESULTS

Similarly to our observation in Section 3.2 for a RESNET-50, a fully shared VIT-B16 trained with the co-training loss Eq. 1 fails to retain any robustness. Therefore, we first investigate various adapters for VIT-B16 to find an efficient training setting in Section 5.1. Then we study *adversarial model soups* with adapters in Section 5.2 and finally show that training with adapters generalizes to other datasets and threat models.

## 5.1 FINDING AN EFFICIENT SETTING

**Choice of adapter.** Using adapters increases the number of parameters as the layers which we choose as adapters have twice as many parameters: one set of parameters for clean images and another for adversarial images. Hence, to avoid increasing the network memory footprint too heavily, we restrict our adapters study to layers with few parameters, thus excluding self-attention (Vaswani et al., 2017) layers and MLP layers. This leaves the options of having dual embedder, positional embedding, normalization layers or classification token; among them, the classification token has by far the least amount of parameters, 49-770× fewer than the other candidates (see details in Table 1). We must still verify that so few parameters are enough to preserve the advantages of the AdvProp training scheme. Hence, we train a model for each type of adapter and compare them with two models without adapters, one trained with nominal training and the other with adversarial training. We observe in Table 1 that by using two classification tokens as adapters, which means only 768 extra parameters out of 86M, we reach 83.56% clean accuracy on IMAGENET, which is an improvement of +0.92% over standard training. Moreover, we obtain a robust accuracy of 49.87% in the *robust mode*, which is close to the robust accuracy given by adversarial training. Notably, we see that adapting other layers with more parameters such as all LayerNorm scales and offsets results in similar performances in both *clean* and *robust modes*. This indicates that *(i)* it is not necessary to split the normalization layers to reproduce the effect of AdvProp, and *(ii)* even a very small amount of dual parameters provide sufficient expressiveness to adapt the shared portion of the network to the two modes. Therefore, in the rest of the manuscript we focus on dual classification tokens as it requires the smallest number of extra parameters.

**Number of attack steps.** As the results in Table 1 were obtained with $PGD^2$, we check if we can reduce the number of attack steps to be more computationally efficient. In Table 2, we report the results for two one-step methods: N-FGSM by de Jorge et al. (2022) and FAST-AT by Wong et al. (2020). If we use the step sizes recommended in the corresponding papers, both methods suffer from *catastrophic overfitting* (Wong et al., 2020) (illustrated in Figure 6 in the appendix) and therefore have no robustness at all. In Table 2 we avoid such *catastrophic overfitting* by reducing the step sizes to $\epsilon$ and $0.75\epsilon$ for FAST-AT and N-FGSM respectively and we observe that both methods perform more than 1% worse in robust accuracy than $PGD^2$. We also increase the number of attack steps to 5 with $PGD^5$. We notice a small improvement over $PGD^2$ of 0.4% in robust accuracy while the clean accuracy is on par. Hence, $PGD^2$ seems to be a good compromise between efficiency and classification performance.

**Table 1: Influence of the adapter type.** We report the clean and robust accuracies in *clean* and *robust mode* on IMAGENET of networks trained with different layer types as adapters. We compare them with networks without adapters trained with nominal training and adversarial training. We recall that VIT-B16 has 86M parameters.

| SETUP | ADAPTER | CLEAN MODE | | ROBUST MODE | |
|---|---|---|---|---|---|
| | # params | CLEAN | ROBUST | CLEAN | ROBUST |
| **BASELINES** | | | | | |
| Nominal Training | 0 | 82.64% | 0% | - | - |
| Adversarial Training | 0 | - | - | 76.88% | 56.19% |
| Co-training (Eq. 1) | 0 | 82.37% | 0% | - | - |
| AdvProp | 38k | 83.39% | 0.02% | 80.96% | 32.97% |
| **VARIOUS ADAPTERS** | | | | | |
| Embedder | 591k | 83.51% | 0.01% | 77.68% | 50.69% |
| Positional embedding | 151k | 83.65% | 0.02% | 77.75% | 49.99% |
| All LN's scales/offsets | 38k | 83.63% | 0% | 77.65% | 50.02% |
| Classification token | **0.8k** | 83.56% | 0.01% | 77.69% | 49.87% |

**Table 2: Influence of the number of attack steps.** We report the clean (in *clean* mode) and robust accuracy (in *robust* mode) on IMAGENET training with various number of attack steps.

| ATTACK | # steps | CLEAN | ROBUST |
|---|---|---|---|
| FAST-AT | 1 | 83.62% | 48.49% |
| N-FGSM | 1 | 83.54% | 47.76% |
| $PGD^2$ | 2 | 83.56% | 49.87% |
| $PGD^5$ | 5 | 83.60% | 50.27% |

**Weighting the co-training loss.** In the co-training loss in Eq. 1, the $\alpha$ hyperparameter controls how much the loss is weighted towards clean or adversarial samples. For example, setting $\alpha = 0$ means we train solely on adversarial samples. In Figure 3, where we evaluate several values for $\alpha$ (dividing the range between 0 and 1 into intervals of length 0.1), we notice that only the values between $\alpha = 0$ and $\alpha = 0.4$ form a Pareto front that strictly dominates the other intervals. Indeed, between $\alpha = 1$ and $\alpha = 0.4$, decreasing $\alpha$ leads to better performance both in *clean* and *robust modes*. In fact,

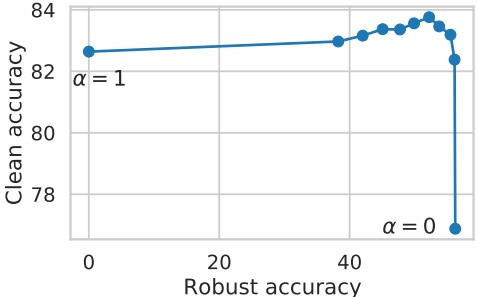 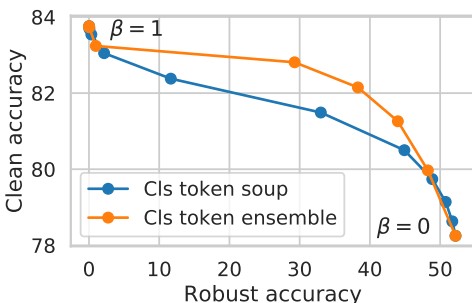

**Figure 3: Co-training loss weighting.** We report the clean accuracy (in *clean* mode) and robust accuracy (in *robust* mode) on IMAGENET when training with various weightings of the co-training loss with adapters Eq. 2. We recall that $\alpha = 1$ corresponds to pure nominal training and $\alpha = 0$ to adversarial training.

**Figure 4: Soup vs ensemble.** We report the clean and robust accuracy (single mode for both) on IMAGENET for various *model soups* and ensembles of a same network trained with classification token adapters.

setting $\alpha = 0.4$ leads to 83.76% clean accuracy (in *clean mode*) and 52.19% robust accuracy (in *robust mode*) which are both better than the values obtained in Table 1 with $\alpha = 0.5$. In Figure 7 (in the appendix), we visualize the filters of the embedder when training with various values of $\alpha$. We observe that for $\alpha = 0.2$ and for $\alpha = 0.8$ the filters look quite similar to the filters learned with adversarial training ($\alpha = 0$) and nominal training ($\alpha = 1$), respectively. Interestingly, filters learned with $\alpha = 0.4$ and $\alpha = 0.6$ are not the simple combination of nominal and adversarial filters but rather new visually distinct filters. This indicates that co-training on clean and adversarial samples can lead to a new hybrid representation for the shared layers compared to nominal and adversarial training.

**Robustness to stronger attacks.** For completeness we further test the robustness of a subset of our models with a mixture of AUTOATTACK (Croce & Hein, 2020) and MULTITARGETED (Gowal et al., 2019), denoted by AA+MT. Pure adversarial training, which obtains 56.19% robust accuracy against PGD$^{40}$ (Table 1), reaches 54.15% robust accuracy against AA+MT. This is a new state-of-the-art robust accuracy on IMAGENET, improving by +6.55% over the 47.60% reported by Debenedetti et al. (2022). While Debenedetti et al. (2022) advocate for weak data augmentation for training robust VIT, our training procedure follows He et al. (2022) and contains heavy augmentations (see appendix): we conclude that large models still benefit from strong data augmentations even with adversarial training. Finally, the *robust mode* of the model co-trained with $\alpha = 0.4$ in the previous paragraph reaches 49.55% robust accuracy against AA+MT, which still surpasses the prior art and preserves competitive robust performance.

## 5.2 EVALUATING MODEL SOUPS

**Adapters enable *adversarial model soups*.** One downside of using adapters is that one needs to know if for an input image the network should be put in *clean* or *robust mode*. This motivates *adversarial model soups* which allow to create a single model performing well both in clean and robust accuracy. First, if we independently train two VIT-B16, one nominally and the other adversarially, and then try to perform *model soups* on them, we notice in Table 9 (in the appendix) that both robust and clean accuracies drop immediately when the weighting factor $\beta$ between parameters is not equal to 0 or 1. We evaluate various *model soups* with the models of Table 1, meaning that the parameters specific to the clean and robust domain are averaged with weight $\beta$ to obtain a single classifier. We notice in Figure 9 (in the appendix) that *adversarial model soups* work equally well with the various types of adapters, where sliding the value of $\beta$ allows to smoothly trade-off clean accuracy for robustness. This validates our hypothesis that adapters enable *model soups*.

**Soup or ensemble.** In Figure 4 we compare the classification performance of *adversarial model soups* and ensembles obtained by linear combination of the *clean* and *robust modes* at the probability prediction level. We notice that ensembling produces a better Pareto front than *adversarial model soup* but ensembles, with their two forward passes, require twice the compute of *model soups*. Hence,

**Figure 5: Soups for IMAGENET variants.** We report the accuracy on IMAGENET variants for *adversarial model soups*. The x-axis corresponds to the interpolation weighting factor $\beta$ where $\beta = 1$ boils down to the *clean mode* and $\beta = 0$ to the *robust mode*. Red (blue) means better (worse) than the *clean mode* (in grey).

*adversarial model soups* allow to choose the trade-off between clean and robust accuracy with performance close to ensembling while only requiring the same compute as a single network.

**Extrapolation.** For the anecdote, we experiment with *adversarial model soups* for extrapolation with values of the weighting factor $\beta$ above 1 and below 0. Interestingly, we observe that setting $\beta = 1.05$ leads to 83.81% clean accuracy which is better than the 83.76% obtained in the *clean mode*. Similarly, setting $\beta = -0.05$ leads to 52.26% robust accuracy which is slightly better than the 52.19% obtained in the *robust mode*. Hence, it appears that *adversarial model soups* do not need to be restricted to interpolation.

**Soups for IMAGENET variants.** As *adversarial model soups* allow to create models with chosen trade-off between clean and robust accuracy, we might expect that such models perform better than nominal ones when distribution shifts occur. For example, Kireev et al. (2021) showed that adversarial training can even help with common corruptions when specifically tuned for such task (note that they use smaller datasets than IMAGENET). We then compute the accuracy of *adversarial model soups* with varying $\beta$ on IMAGENET variants (results in Figure 5): while half of the best performance are obtained with the *clean* classification token, for some variants such as IMAGENET-R, IMAGENET-C and IMAGENET-SKETCH the best results are obtained with intermediate tokens. Hence, *adversarial model soups* can be used to reach a compromise between IMAGENET variants to get the best average performance. Here $\beta = 0.9$ yields the best mean accuracy 61.23%. In Table 3, we notice that this *adversarial model soup* improves the mean accuracy by +4.00% over a fine-tuned Masked Autoencoder (MAE-B16) checkpoint from He et al. (2022) and by +2.37% over Pyramid-AT from Herrmann et al. (2022). It also improves by +2.24% over the best performing ensemble of two networks trained independently with nominal and adversarial training respectively.

**Table 3: IMAGENET variants results.** We report the *adversarial model soup* achieving the best average accuracy over the eight variants and we compare it against several baselines. All models are trained on IMAGENET and use the same VIT-B16 architecture.

| SETUP | COMPUTE | IMAGENET | IN-REAL | IN-V2 | IN-A | IN-R | IN-SKETCH | CONFLICT | IN-C | MEAN |
|---|---|---|---|---|---|---|---|---|---|---|
| **BASELINES** | | | | | | | | | | |
| Nominal training | ×1 | 82.64% | 87.33% | 71.42% | 28.03% | 47.94% | 34.43% | 30.47% | 64.45% | 55.84% |
| Adversarial training | ×1 | 76.88% | 83.91% | 64.81% | 12.35% | 55.76% | 40.11% | **59.45%** | 55.44% | 56.09% |
| Fine-tuned MAE-B16 | ×1 | 83.10% | 88.02% | 72.80% | 37.92% | 49.30% | 35.69% | 27.81% | 63.23% | 57.23% |
| Pyramid-AT (fully shared network) | ×1 | 83.14% | 87.82% | 72.53% | 32.72% | 51.78% | 38.60% | 37.27% | 67.01% | 58.86% |
| Independent networks ensemble | ×2 | 82.86% | 87.78% | 71.73% | 25.99% | 54.20% | 37.33% | 46.41% | 65.61% | 58.99% |
| **WITH ADAPTERS** | | | | | | | | | | |
| Classification token soup | ×1 | **83.69%** | 88.50% | **73.48%** | **38.23%** | 54.74% | 41.17% | 40.00% | **70.07%** | 61.23% |
| Classification token ensemble | ×2 | 83.62% | **88.58%** | 73.36% | 35.05% | **56.32%** | **41.36%** | 49.14% | 70.04% | **62.18%** |

**Table 4: Comparing threat models.** We optimize the co-training loss with adversarial examples coming from various attacks. For each run, we select the *adversarial model soup* achieving the best mean performance on IMAGENET variants, which we report in the table. We set the co-training loss weighting factor $\alpha = 0.5$.

| ATTACK | IMAGENET | IN-REAL | IN-V2 | IN-A | IN-R | IN-SKETCH | CONFLICT | IN-C | MEAN |
|---|---|---|---|---|---|---|---|---|---|
| Nominal training | 82.64% | 87.33% | 71.42% | 28.03% | 47.94% | 34.43% | 30.47% | 64.45% | 55.84% |
| $\ell_\infty$ (untargeted) | 83.36% | 88.31% | 72.73% | 34.81% | 54.15% | 40.37% | 41.25% | 69.54% | **60.56%** |
| $\ell_2$ (untargeted) | 83.35% | 88.12% | 72.82% | 35.17% | 52.59% | 39.34% | 38.75% | 68.99% | 59.89% |
| $\ell_\infty$ (targeted) | 83.21% | 87.97% | 72.95% | 33.87% | 52.90% | 38.70% | 39.06% | 68.49% | 59.64% |
| Pyramid-$\ell_\infty$ | 83.62% | 88.20% | 73.36% | 35.15% | 53.94% | 39.40% | 41.88% | 68.39% | 60.49% |

**Table 5: Evaluating on other datasets.** We report the clean accuracy (in the *clean mode*) when training on various datasets with several perturbation radii. We compare it with nominal training on the same architecture.

| RADIUS | MNIST | CIFAR-10 | CIFAR-100 | SVHN | SUN-397 | RESISC-45 | DMLAB |
|---|---|---|---|---|---|---|---|
| Nominal | 99.48% | 94.63% | 79.21% | 97.88% | 65.24% | 94.03% | 55.66% |
| 1/255 | 99.56% | 96.68% | 80.88% | **98.38%** | 67.08% | 95.09% | **58.94%** |
| 2/255 | 99.67% | **97.07%** | 81.02% | 98.14% | 67.27% | **95.20%** | 57.99% |
| 4/255 | 99.68% | 97.00% | 80.97% | 97.83% | **67.32%** | 94.86% | 57.81% |
| 8/255 | **99.71%** | 96.34% | **81.27%** | 96.18% | 65.97% | 94.12% | 53.64% |

## 5.3 EVALUATING ON OTHER THREAT MODELS AND DATASETS

**Evaluating other threat models.** IMAGENET variants are also a good benchmark to compare different types of adversarial attack to generate the perturbations for the co-training loss in Eq. 2: untargeted $\ell_\infty$-bounded perturbations with budget $\epsilon = 4/255$ (our standard setup), untargeted $\ell_2$-bounded with $\epsilon \in \{1, 2, 4, 8\}$, targeted (random target class as in Xie et al., 2019a) $\ell_\infty$-bounded with $\epsilon \in \{4/255, 8/255, 12/255\}$, and the Pyramid attack proposed by Herrmann et al. (2022). In Table 4, we select the best *adversarial model soups* after training with each method a VIT-B16 with dual classification tokens, and report its results on all variants. We see that the clean accuracy on the IMAGENET validation set improves in all cases compared to standard training. Moreover, although the best performing attack varies across variants, we notice that the untargeted $\ell_\infty$ attack achieves the best average accuracy.

**Evaluating on other datasets.** We further test the effect of using the co-training loss with the classification token as adapter on other datasets. In Table 5, we see that our training procedure provides a consistent performance boost in clean accuracy compared to nominal training on MNIST (LeCun et al., 2010), CIFAR-10, CIFAR-100 (Krizhevsky et al., 2014), SVHN (Netzer et al., 2011), SUN-397 (Xiao et al., 2010), RESISC-45 (Cheng et al., 2017) and DMLAB (Beattie et al., 2016). This shows that our method generalizes well across datasets and can help regularize Vision Transformers on these smaller datasets, where they are known to perform worse compared to CNNs without pre-training (Zhang et al., 2021). In Appendix C, we also demonstrate that models pre-trained with co-training on IMAGENET yield significantly better classification results when fine-tuning nominally on small datasets compared to fine-tuning from nominally and adversarially pre-trained models.

## 6 CONCLUSION

In this work we have shown that adapters with a few hundreds of domain specific parameters are sufficient to switch between models with radically different behaviors. In fact, just replacing the classification token of a VIT can turn a classifier with SOTA nominal accuracy and no adversarial robustness into another one with robust accuracy close to that achieved with standard adversarial training. Moreover, merging the adapters allows to smoothly transition between the two modes, finding classifiers (i.e. our *adversarial model soups*) with better performance on distribution shifts. These observations open up new interesting directions for future work to explore how to take advantage of the regularizing effect of adversarial training and whether it is possible to combine via soups other types of models.

## ACKNOWLEDGEMENTS

We are grateful to Evan Shelhamer for reviewing the drafts of the paper and his literature comments, to Olivia Wiles, Florian Stimberg, Taylan Cemgil and others at DeepMind for helpful conversations and feedback on the project.

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

## A  MORE EXPERIMENTAL DETAILS

**Training details.**    In this manuscript we train VIT-B16 models using the training pipeline proposed in He et al. (2022). The model is optimized for 300 epochs using the AdamW optimizer (Loshchilov & Hutter, 2017) with momenta $\beta_1 = 0.9$, $\beta_2 = 0.95$, with a weight decay of 0.3 and a cosine learning rate decay with base learning rate 1e-4 and linear ramp-up of 20 epochs. The batch size is set to 4096 and we scale the learning rates using the linear scaling rule of Goyal et al. (2017). We optimize the standard cross-entropy loss and we use a label smoothing of 0.1. We apply stochastic depth (Huang et al., 2016) with base value 0.1 and with a dropping probability linearly increasing with depth. Regarding data augmentation, we use random crops resized to $224 \times 224$ images, mixup (Zhang et al., 2018), CutMix (Yun et al., 2019) and RandAugment (Cubuk et al., 2020) with two layers, magnitude 9 and a random probability of 0.5. We note that our implementation of RandAugment is based on the version found in the *timm* library (Wightman, 2019). We also use exponential moving average with momentum 0.9999. For RESNET-50 we keep the same training scheme used for VIT-B16, and the standard architecture, except for combining GroupNorm with Weight Standardization in all convolutional layers following Kolesnikov et al. (2020). For the DualParams BatchNorm version we fix the *robust* branch to always use the accumulated statistics rather then the batch ones.

**Training on smaller datasets.**    When training from scratch on smaller datasets in Section 5.3, we optimize the smaller VIT-S with a batch size of 1024 and a base learning rate of 2e-4. For datasets with small image resolution such as CIFAR-10, we do not rescale the images to $224 \times 224$ but we use a patch size of 4 and a stride of 2 to get enough vision tokens.

**Attack details.**    For $PGD^2$ and $PGD^5$ we use a gradient descent update with a fixed step size of 2.5/255 and 1/255 respectively. For $PGD^{40}$ we change the optimizer to Adam with step-size 0.1 decayed by $10 \times$ at steps 20 and 30. Regarding one step attacks, we use a step size of 6/255 and initialization radius of 8/255 for N-FGSM and a step size of 5/255 for Fast-AT.

## B  VISUALIZING FILTERS

**Visualization procedure.**    We visualize the embedding layer by first standardizing the weights to have zero mean and unit variance. We then extract the first 28 principal components. Finally we reshape them to $16 \times 16 \times 3$ images and rescale them to have their values between 0 and 255 such as to display these components as RGB images.

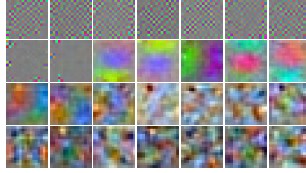
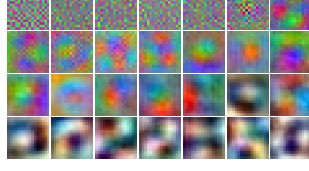
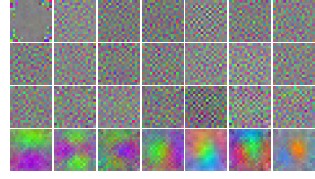

(a) $PGD^2$ but no adapters          (b) Fast-AT with $1.25\epsilon$ step size          (c) N-FGSM with $1.5\epsilon$ step size

**Figure 6:** First 28 principal components of the embedding filters of VIT-B16. In panel (a), a fully shared model (no adapters) is co-trained on clean samples and adversarial samples coming $PGD^2$. In panels (b) and (c), models with classification token adapters are adversarially co-trained with adversarial samples coming from 1 step attacks Fast-AT and N-FGSM when using their recommended step sizes. In these three cases, we observe *catastrophic robust overfitting* with no robustness at the end of training. We observe that there are degenerate filters among the first principal components. For panels (b) and (c), robust overfitting can be avoided by reducing their step sizes.

## C  TRANSFER LEARNING

**Training details.**    For completeness we evaluate the transfer learning performance of the VIT-B16 pre-trained on IMAGENET by co-training on clean and adversarial samples. We choose the

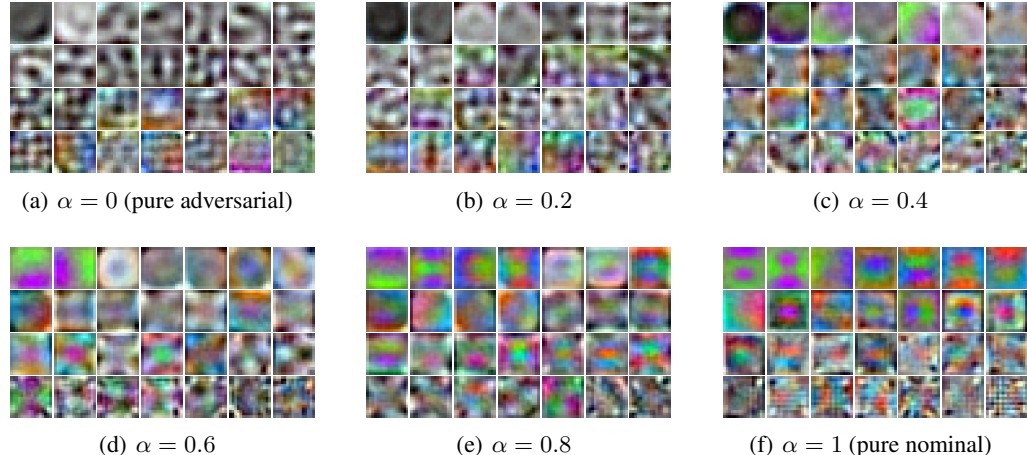

**Figure 7:** First 28 principal components of the embedding filters of VIT-B16 with classification token adapter trained with various weightings $\alpha$ of the co-training loss Eq. 1. We recall that $\alpha = 1$ corresponds to pure nominal training and $\alpha = 0$ to adversarial training. Filters learned with $\alpha = 0$ focus more on intensity variations (black and white patterns) whereas filters learned with $\alpha = 0$ are more color based. Filters with intermediate values of $\alpha$ exhibit circle and star patterns which are not present for $\alpha = 0$ or $\alpha = 1$.

model trained with classification token adapter and co-training coefficient $\alpha = 0.4$, which we fine-tune nominally on CIFAR-10, CIFAR-100, SUN-397, RESISC-45 and DMLAB using SGD with momentum 0.9, a batch size of 512, gradient clipping at global norm 1 and no weight decay. We optimize the standard cross-entropy loss and we use a label smoothing of 0.1. For simplicity, we use the same training schedule for all the datasets: a total of 10k training steps and a base learning rate of 0.01 attained after a linear ramp-up of 500 steps followed by a cosine decay. Regarding data pre-processing, we simply rescale the images to $224 \times 224$ resolution without preserving aspect ratio and we apply random horizontal flipping as data augmentation. Finally, we use exponential moving average with momentum 0.999.

**Fine-tuning results.** As the network was pre-trained with classification token adapter, we have several possibilities for initializing the classification token before fine-tuning: adversarial token, clean token and *model soups* interpolating between these two tokens. For comparison, we also fine-tune two VIT-B16 pre-trained on IMAGENET with nominal and adversarial training respectively. We report the results in Table 6 where we evaluate several fine-tuning strategies: fine-tuning *(i)* the classifier head, *(ii)* the classifier head and the classification token and *(iii)* all the weights. First, we observe that fine-tuning both the classification token and the classifier head brings only a small improvement (from 79.27% to 80.70% for the best average accuracy) over fine-tuning the classifier head alone. Fine-tuning all the weights is the best strategy as it reaches 88.40% average accuracy. Second, we observe that initializing the classification token with the adversarial token performs consistently better than with the clean token when fine-tuning all the weights. Finally, co-training as pre-training is significantly better than nominal and adversarial pre-training as fine-tuning from a co-trained model reaches 88.40% average accuracy, a +1.05% improvement over nominal and adversarial pre-training.

## D   ACCURACY LANDSCAPE

In our case, *model soups* are obtained by linear interpolation (or extrapolation) of the adversarial and clean tokens. We notice that the clean and adversarial tokens are almost orthogonal ($\cos(\phi_{\text{clean}}, \phi_{\text{adv}}) = 0.14$), so we can extend our study beyond *model soups* by taking linear combinations of the two tokens $\beta_1 \phi_{\text{clean}} + \beta_2 \phi_{\text{adv}}$. By taking a sweep over the $\beta_1$ and $\beta_2$ coefficients, we obtain in Figure 8 the clean and robust accuracy landscapes in the plane defined by the two tokens and where the diagonal corresponds to the *model soups*. We observe that the main direction of change for the clean and robust accuracies is the *model soups* diagonal (top left to bottom right). We can clearly see the trade-off in clean/robust accuracy, but also there seems to be a compromise

**Table 6: Co-training as pre-training.** We compare the transfer learning performance of a model pre-trained using co-training to models pre-trained with nominal and adversarial training. We evaluate various fine-tuning strategies on several datasets (headers in green) and we report the average over datasets in the last rows (orange header). We also assess several initializations for the classification token before fine-tuning: adversarial token, clean token and *model soups* between these two tokens with various weightings $\beta$. All models are pre-trained on IMAGENET and use the same VIT-B16 architecture during fine-tuning.

| SETUP | BASELINES | | FROM CO-TRAINED NET | | | | |
|---|---|---|---|---|---|---|---|
| | Nominal | Adversarial | Robust mode | $\beta = 0.25$ | $\beta = 0.5$ | $\beta = 0.75$ | Clean mode |
| CIFAR-10 | | | | | | | |
| Fine-tune head | 96.07% | 90.95% | 90.28% | 91.17% | 93.61% | 96.50% | **97.15%** |
| Fine-tune head + cls token | 96.62% | 92.76% | 97.73% | 97.70% | 97.77% | 97.82% | **97.84%** |
| Fine-tune all | 98.68% | 98.96% | **99.09%** | 99.03% | 99.01% | 99.05% | 99.03% |
| CIFAR-100 | | | | | | | |
| Fine-tune head | 83.30% | 73.80% | 71.94% | 73.52% | 77.78% | 83.99% | **85.47%** |
| Fine-tune head + cls token | 84.59% | 76.79% | 87.26% | 87.49% | **87.55%** | 87.45% | 87.43% |
| Fine-tune all | 91.18% | 91.74% | 92.37% | 92.23% | 92.32% | **92.41%** | 92.29% |
| SUN-397 | | | | | | | |
| Fine-tune head | 72.70% | 65.62% | 65.93% | 67.02% | 70.19% | 73.00% | **73.47%** |
| Fine-tune head + cls token | 73.05% | 67.21% | 73.99% | 74.14% | **74.19%** | 74.12% | 74.15% |
| Fine-tune all | 76.48% | 75.66% | **77.87%** | 77.75% | 77.74% | 77.67% | 77.72% |
| RESISC-45 | | | | | | | |
| Fine-tune head | **91.69%** | 86.70% | 86.54% | 87.37% | 89.64% | 90.58% | 91.12% |
| Fine-tune head + cls token | **91.95%** | 87.52% | 91.04% | 91.07% | 91.04% | 91.49% | 91.23% |
| Fine-tune all | 96.78% | 96.14% | **97.07%** | 96.72% | 96.88% | **97.07%** | 96.80% |
| DMLAB | | | | | | | |
| Fine-tune head | 50.02% | **50.11%** | 48.58% | 48.60% | 49.08% | 49.07% | 49.16% |
| Fine-tune head + cls token | 50.91% | 51.53% | 50.81% | 51.79% | 52.47% | **52.64%** | 52.41% |
| Fine-tune all | 73.65% | 73.93% | 75.61% | 75.66% | **75.74%** | 75.35% | 75.58% |
| AVERAGE | | | | | | | |
| Fine-tune head | 78.76% | 73.44% | 72.65% | 73.54% | 76.06% | 78.63% | **79.27%** |
| Fine-tune head + cls token | 79.42% | 75.16% | 80.17% | 80.44% | 80.60% | **80.70%** | 80.61% |
| Fine-tune all | 87.35% | 87.29% | **88.40%** | 88.28% | 88.34% | 88.31% | 88.28% |

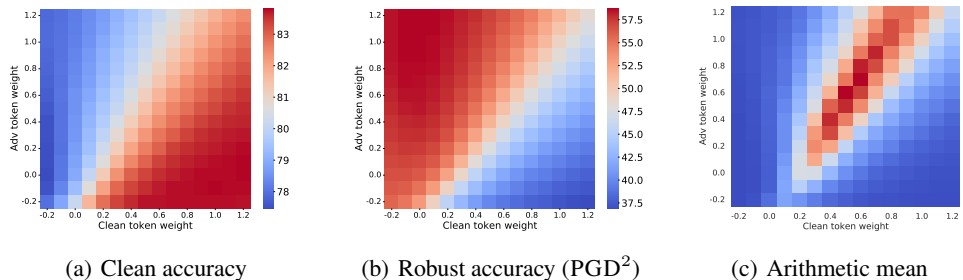

(a) Clean accuracy     (b) Robust accuracy (PGD$^2$)     (c) Arithmetic mean

**Figure 8: Linear combination of tokens.** We report the clean accuracy (panel (a)) and robust accuracy against PGD$^2$ (panel (b)) on IMAGENET for various linear combinations of the clean and adversarial tokens. *Model soups*, which are linear interpolation (and extrapolation) between these two tokens, are on the diagonal from top left to bottom right. Panel (c) shows the arithmetic mean between the normalized (with min/max rescaling) clean and robust accuracies (red means higher mean accuracy).

region between clean and robust accuracy as the other diagonal (from bottom left to top right) is visually distinct for clean and robust accuracy. In panel (c) of Figure 8, we plot the arithmetic mean between the normalized (with min/max rescaling) clean and robust accuracies. We observe that the best compromises between clean and robust accuracy have a stronger adversarial token weight than the clean token weight.

# E  LIMITATIONS AND FUTURE WORK

We have empirically shown that co-training a fully shared VIT does not retain any robustness whereas having two classification tokens specific to the clean and adversarial images is enough to get competitive performance both in clean and robust accuracy. However, we leave to future work the theoretical explanation on why this small architecture change (adding only 768 parameters) results in such a gap in performance. Similarly, beyond our intuition that parameter sharing when using adapters makes *model soups* possible, we cannot support our empirical results with theory and leave it to future work. Another direction for future work is the automatic selection of the right soup for each sample which could be inspired by automatic selection modules like in Lo & Patel (2021).

# F  ADDITIONAL TABLES AND FIGURE

In the following we present additional tables and figures of results described in the main part but omitted above because of space limits.

**Table 7: Model soups for several weightings $\beta$.** We report the accuracy for various *model soups* on IMAGENET variants by interpolating between the clean and adversarial tokens with several weightings $\beta$. $\beta = 1$ boils down to the *clean mode* and $\beta = 0$ to the *robust mode*.

| $\beta$ | IMAGENET | IN-REAL | IN-V2 | IN-A | IN-R | IN-SKETCH | CONFLICT | IN-C | MEAN |
|---|---|---|---|---|---|---|---|---|---|
| 0 | 78.25% | 84.73% | 66.04% | 13.39% | 55.23% | 39.62% | **56.48%** | 56.67% | 56.30% |
| 0.1 | 78.63% | 85.03% | 66.49% | 14.00% | 55.31% | 39.79% | 55.70% | 57.26% | 56.53% |
| 0.2 | 79.14% | 85.40% | 67.13% | 15.11% | 55.23% | 40.00% | 54.37% | 58.23% | 56.83% |
| 0.3 | 79.74% | 85.89% | 67.95% | 16.96% | 55.40% | 40.26% | 53.36% | 59.73% | 57.41% |
| 0.4 | 80.50% | 86.47% | 68.91% | 19.60% | 55.55% | 40.47% | 51.09% | 61.91% | 58.06% |
| 0.5 | 81.49% | 87.18% | 70.21% | 23.56% | **55.64%** | 40.70% | 49.14% | 64.74% | 59.08% |
| 0.6 | 82.38% | 87.76% | 71.43% | 28.63% | 55.58% | 40.98% | 46.88% | 67.52% | 60.15% |
| 0.7 | 83.05% | 88.26% | 72.41% | 33.72% | 55.43% | 41.19% | 44.38% | 69.27% | 60.96% |
| 0.8 | 83.54% | 88.46% | 72.94% | 36.79% | 55.07% | **41.22%** | 41.64% | 69.97% | 61.20% |
| 0.9 | 83.69% | 88.50% | 73.48% | 38.23% | 54.74% | 41.17% | 40.00% | **70.07%** | **61.23%** |
| 1 | **83.76%** | **88.52%** | **73.53%** | **38.37%** | 54.43% | 41.15% | 39.77% | 69.92% | 61.18% |

**Table 8: Ensembles for several weightings $\beta$.** We report the accuracy on IMAGENET variants for various ensembles obtained by averaging the probability predictions of the *clean and robust modes* of a VIT-B16 with classification token adapter. Here $\beta$ is the weight used to average the probability predictions. $\beta = 1$ boils down to the *clean mode* and $\beta = 0$ to the *robust mode*.

| $\beta$ | IMAGENET | IN-REAL | IN-V2 | IN-A | IN-R | IN-SKETCH | CONFLICT | IN-C | MEAN |
|---|---|---|---|---|---|---|---|---|---|
| 0 | 78.25% | 84.73% | 66.04% | 13.39% | 55.23% | 39.62% | **56.48%** | 56.67% | 56.30% |
| 0.1 | 79.97% | 86.10% | 68.33% | 21.23% | 56.26% | 40.15% | 56.41% | 64.14% | 59.07% |
| 0.2 | 81.26% | 87.11% | 70.03% | 25.13% | 56.56% | 40.63% | 55.08% | 67.09% | 60.36% |
| 0.3 | 82.15% | 87.74% | 71.24% | 27.72% | 56.63% | 40.95% | 53.67% | 68.57% | 61.08% |
| 0.4 | 82.81% | 88.16% | 72.09% | 29.96% | **56.68%** | 41.11% | 52.50% | 69.36% | 61.58% |
| 0.5 | 83.24% | 88.39% | 72.88% | 32.13% | 56.65% | 41.28% | 51.41% | 69.79% | 61.97% |
| 0.6 | 83.47% | 88.51% | 73.28% | 33.64% | 56.46% | 41.35% | 50.23% | 69.98% | 62.12% |
| 0.7 | 83.62% | 88.58% | 73.36% | 35.05% | 56.32% | **41.36%** | 49.14% | **70.04%** | **62.18%** |
| 0.8 | 83.68% | 88.58% | 73.37% | 36.27% | 56.03% | 41.35% | 46.64% | **70.04%** | 61.99% |
| 0.9 | 83.73% | 88.57% | 73.47% | 37.25% | 55.62% | 41.28% | 44.61% | 69.99% | 61.81% |
| 1 | **83.76%** | **88.52%** | **73.53%** | **38.37%** | 54.43% | 41.15% | 39.77% | 69.92% | 61.18% |

**Table 9:** *Model soups* **on independent models vs on a network co-trained with adapters.** We report the clean and robust accuracy for *model soups* with various weightings $\beta$ of either (1) two independently trained VIT-B16, one nominally and the other adversarially in the first group of columns or (2) a single co-trained model with classification token adapters in the last two columns. When souping the two independent models, $\beta = 0$ boils down to the model trained adversarially and $\beta = 1$ to the model trained nominally. We notice for these soups between the two independent models that both robust and clean accuracies drop immediately when the weighting factor $\beta$ between parameters is not equal to 0 or 1. Regarding the soups between the clean and adversarial tokens (whose numbers correspond to the blue curve in Figure 4), we notice that the *adversarial model soups* achieve a smooth transition between the clean and robust modes.

| $\beta$ | INDEPENDENT NETS | | ADAPTERS | |
|---|---|---|---|---|
| | CLEAN | ROBUST | CLEAN | ROBUST |
| 0 | 76.88% | 56.19% | 78.25% | 52.19% |
| 0.1 | 28.72% | 8.56% | 78.63% | 51.73% |
| 0.2 | 1.38% | 0.06% | 79.14% | 50.79% |
| 0.3 | 0.21% | 0.00% | 79.74% | 48.88% |
| 0.4 | 0.16% | 0.00% | 80.50% | 44.91% |
| 0.5 | 0.14% | 0.01% | 81.49% | 33.01% |
| 0.6 | 0.10% | 0.01% | 82.38% | 11.65% |
| 0.7 | 1.72% | 0.09% | 83.05% | 2.16% |
| 0.8 | 43.25% | 0.14% | 83.54% | 0.34% |
| 0.9 | 78.56% | 0.01% | 83.69% | 0.06% |
| 1 | 82.64% | 0.00% | 83.76% | 0.02% |

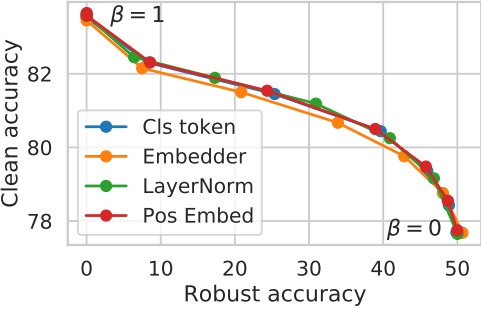

**Figure 9: Comparing soups for various adapters.** We report the clean and robust accuracy (single mode for both) on IMAGENET for *model soups* of networks trained with various types of adapters.

**Table 10: Robustness to stronger attacks.** We report the clean accuracy (in *clean* mode) and robust accuracy (in *robust* mode) against $PGD^{40}$ and against the stronger AA+MT on IMAGENET for various co-training loss weightings $\alpha$. We report the accuracies for $\alpha$ ranging from $\alpha = 0$ to $\alpha = 0.4$ as we recall from Figure 3 that it corresponds to the Pareto front when compromising between clean and robust accuracy.

| $\alpha$ | CLEAN | ROBUST ($PGD^{40}$) | ROBUST (AA+MT) |
|---|---|---|---|
| 0 | 76.88% | 56.19% | 54.15% |
| 0.1 | 82.38% | 56.09% | 53.98% |
| 0.2 | 83.19% | 55.46% | 53.24% |
| 0.3 | 83.46% | 53.73% | 51.37% |
| 0.4 | 83.76% | 52.19% | 49.55% |

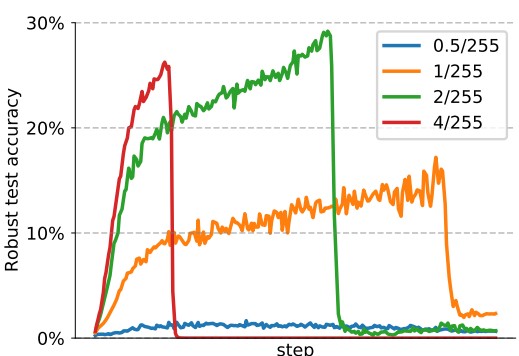

**Figure 10: Catastrophic robust overfitting when co-training without adapters.** We report the robust accuracy against $\ell_\infty$ perturbations of size $\epsilon = 4/255$ over training steps when co-training with various perturbation radii a VIT-B16 without adapters. We observe that only $\epsilon_\infty = 0.5/255$ does not suffer from a collapse in robust accuracy during training.

