# OpenReview forum: "Revisiting adapters with adversarial training"
_ICLR.cc/2023/Conference — ICLR 2023 notable top 25%_

### Official Review · Reviewer_NrHB · 2022-10-18

**Confidence:** 4
**Clarity, Quality, Novelty And Reproducibility:** See Strength And Weaknesses.
**Correctness:** 4
**Technical Novelty And Significance:** 4
**Empirical Novelty And Significance:** 4
**Recommendation:** 8

**Strength And Weaknesses:**

Strength
+ It is very novel and inspiring to separate tokens of VIT for domain-specific training in the adversarial setting, and implementing the well-known idea (clean and adversarial samples come from different domains) in the new tool (VIT) is interesting and inspiring.
+ The authors also insightfully demonstrate that training with adapters enables model soup, which allows a simple trade-off between robustness and accuracy in adversarial model soup by interpolation between the clean and adversarial token. It frees the burden of retraining the whole model when the balance is to change.
+ Extensive experiments show that the proposed method could achieve SOTA clean and robust performance in different modes. Besides, the results of the distribution shift are also encouraging.
+ The authors also thoroughly analyze the influence of adapters, attack steps, the weighting hyperparameter, extrapolation, etc. The results in Table 9 convincingly necessitate the weight-sharing design.
+ The paper is well-motivated, well-organized, and easy to follow.

Weakness
+ The authors should better illustrate how the adversarial examples are included in clean mode training.
+ It would be better to report the adversarial model soup performance on other datasets, e.g., CIFAR-10, CIFAR-100, along with other models in the RobustBench.
+ The authors should use \citep instead of \cite in some places and maintain a larger font size for tables.
+ Section 3.2 does not seem necessary to be so long. In contrast, the results of the adversarial model soup, Figure 9, are more important to me.
+ Would additional data further increase the performance as in [1,2]?


[1] Improving robustness using generated data, NeurIPS 2021.

[2] Data augmentation can improve robustness, NeurIPS 2021.

**Summary Of The Paper:**

The paper proposes to treat clean and adversarial samples as different domains and co-train them by separate classification tokens of a VIT, which produces adversarial model soup to trade off between clean and robust accuracy by simple interpolation. The authors present a good alternative for advprop with fewer parameters and better robustness.

**Summary Of The Review:**

The paper is very novel and clear with convincing results. The experiments are thorough and sound.

---

> ### Author Response · Authors · 2022-11-11
> **Thank you for the review**
>
> Thank you for the review. Please see answers below.
>
> > The authors should better illustrate how the adversarial examples are included in clean mode training.
>
> As written in Equation 2, adversarial examples are used to train the shared parameters and the parameters of the robust branch $\\phi_\\textrm{adv}$ , but do not influence the parameters of the clean branch $\\phi_\\textrm{clean}$. We have added $\\phi_\\textrm{clean}$ and $\\phi_\\textrm{adv}$ from Equation 2 in Figure 1 to illustrate this point. Maybe we have not understood the remark of the reviewer.
>
> > It would be better to report the adversarial model soup performance on other datasets, e.g., CIFAR-10, CIFAR-100, along with other models in the RobustBench.
>
> In the answer to reviewer VYKg, we provide the adversarial model soup performance on ImageNet for the ResNet-50 with dual BatchNorm of Figure 2. We observe that adversarial model soups work well on this ResNet-50 so they should normally perform well on CIFAR datasets with ResNet architectures.
>
> > The authors should use \citep instead of \cite in some places and maintain a larger font size for tables.
>
> We thank the reviewer for these suggestions and we have modified the manuscript accordingly.
>
> > Would additional data further increase the performance as in [1,2]?
>
> Indeed, instead of using domain-specific parameters for adversarial examples, one could possibly use adapters to co-train on original samples and generated samples to avoid hurting the accuracy on the test set if the generated images are too far from the original data distribution. Similarly, one could also try to improve robust accuracy by co-training on perturbed original samples and perturbed augmented images.

---

> > ### Comment · Reviewer_NrHB · 2022-11-17
> > **Thanks for the response**
> >
> > Thank the authors for a detailed rebuttal. From my perspective, the authors address the concerns well, so I maintain my scores. I personally recommend highlighting and linking the changes in the manuscript to specific reviews for other reviewers to view clearly.

---

### Official Review · Reviewer_Wvkr · 2022-10-21

**Confidence:** 3
**Clarity, Quality, Novelty And Reproducibility:** 1. The baselines **Adversarial Traini…
**Correctness:** 3
**Technical Novelty And Significance:** 3
**Empirical Novelty And Significance:** 3
**Recommendation:** 6

**Strength And Weaknesses:**

Strength
1. Model soups are drawing much attention due to their excellent performance. However, they require a great number of computing resources. The adversarial model soups alleviate the need for storage capacity and computing power and make them practical for mobile devices and independent researchers.
2. Findings in Figure 2 are interesting, which shows that domain-specific normalization layers are enough to boost the robustness.
3. The method is simple, and few hyper-parameters are required.

Weakness
1. AdvProp boosts both accuracy and robustness and doesn’t need more parameters. I think you had better add the results of AdvProp in Table 1.
2. What’s the $\alpha$ used in the baseline **Adversarial Training** in Table 1? Could you do an ablation study here providing the results of different $\alpha$?


**Summary Of The Paper:**

The paper works on the robustness of the neural networks and aims to boost the accuracy on clean ImageNet and different variants with the help of adversarial samples. Inspired by the AdvProp and model soups, the authors propose adversarial model soups, which are trained with adapters through linear combinations of the clean and adversarial tokens. Experiments show model soups can easily strike a balance between clean accuracy and dataset-shifting robustness.


**Summary Of The Review:**

In a word, the paper does a good job for its interesting insights and innovative method. However, it suffers from several problems, like a lack of baselines and clarity.

---

> ### Author Response · Authors · 2022-11-11
> **Thank you for the review**
>
> Thank you for the review. Please see answers below.
>
> > AdvProp boosts both accuracy and robustness and doesn’t need more parameters. I think you had better add the results of AdvProp in Table 1.
>
> We have added AdvProp to Table 1. AdvProp can be regarded as co-training with dual LayerNorm so it uses 38k adapter parameters. We also note that the original implementation of AdvProp uses random targeted attacks. Hence, as the robust evaluation in Table 1 is done with untargeted attacks, AdvProp performs worse in robust accuracy in the robust mode than our other results with adapters. Training with untargeted attacks corresponds to the "All LN’s scales/offsets" row in the same table. One of our contributions is to demonstrate that the effects behind AdvProp are not restricted to normalization layers. Please see answers to reviewer VYKg for more details.
>
> > [What’s the α used in the baseline Adversarial Training in Table 1?][The baselines Adversarial Training and Co-training in Table 1 are confusing. If Co-training uses Eq. 1 as shown in the Table, what’s the difference between them?]
>
> Adversarial training corresponds to training only on adversarial examples, which corresponds to $\\alpha=0$ in Equation 1. The co-training line in Table 1 corresponds to $\\alpha=0.5$ so training the network on both clean and adversarial samples in equal proportion.
>
> > In Section 5.1 (Weighting the co-training loss), you refer to Eq. 1 as the co-training loss. Later, in the caption of Figure 3, you refer the Eq. 2 as the co-training loss. You had better clarify the notions in your paper.
>
> In Equation 1, co-training is done on a network whose parameters are fully shared between clean and adversarial samples. Compared to Equation 1, we introduce in Equation 2 domain-specific parameters $\\phi_\textrm{clean}$ and $\\phi_\textrm{adv}$ which are separately trained on the clean and adversarial samples respectively. We thank the reviewer for the suggestion: we have modified the caption of Figure 3 where we now refer to the Equation 2 as the co-training loss with adapters. We have also modified Figure 1 to include $\\phi_\textrm{clean}$ and $\\phi_\textrm{adv}$ from Equation 2.
>
> > Could you do an ablation study here providing the results of different $\\alpha$ ?
>
> We do this ablation for the dual classification token in Figure 3. Furthermore, we have now added the robustness to stronger attacks for different $\\alpha$ in Table 10 in the appendix. Doing this ablation study for the other adapter types would require too much computation as it would mean co-training 30 models on ImageNet (3 adapter types and 10 values of $\\alpha$).
>
> > Could you fine-tune some adversarial models (with different $\\alpha$ in Eq. 1) from the same pre-trained model, average the parameters just like standard model soups and then test the accuracy and robustness on such a model soup?
>
> We thank the reviewer for this suggestion. Indeed, adversarially pre-training a model, then fine-tuning it nominally and finally souping the original and fine-tuned models is likely to work but this idea opens several other research directions and possible experiments so it is out of the scope of the paper.
>
> > In Fig. 5, the model performs differently on IN-R and Conflict Stimuli. Could you explain it?
>
> ImageNet-R (whose images come from very varied data sources) and even more Conflict Stimuli (whose images should be classified based on shape and not texture) have very different data distribution compared to the original ImageNet training set. In comparison, the other variants are much closer to the original training distribution. Hence, the robust mode is more likely to perform better on these extreme distribution shifts than the clean mode.

---

### Official Review · Reviewer_VYKg · 2022-10-25

**Confidence:** 4
**Correctness:** 3
**Technical Novelty And Significance:** 2
**Empirical Novelty And Significance:** 2
**Recommendation:** 6

**Clarity, Quality, Novelty And Reproducibility:**

The paper is well-written and easy to follow. The idea is interesting but doesn't show advances over the existing AdvProp method.

**Strength And Weaknesses:**

## Strength

1. The method is simple yet effective, it significantly reduces the number of domain-specific parameters compared with AdvProp.
1. The interpolation/extrapolation experiments in Figure 3 are interesting and show the benefit of using the class token as the adapter.

## Weakness

1. Even in the original AdvProp, the number of domain-specific parameters is marginally compared with the number of the whole model, which makes the benefit of further reducing the number of domain-specific parameters less useful.
1. The interpolation, extrapolation, and model soup could also be applied to the AdvProp and the results would be interesting.
1. Using the class token as the adapter so it doesn't work with CNN and transformer without class token, which limits the application of this method.
1. Comparison with AdvProp is not thorough. AdvProp could be regarded as using dual norm as the adapters. The influence of different adapters should be studied.

**Summary Of The Paper:**

This paper shows that separate batch statistics for co-training on clean and adversarial inputs are not necessary. An extremely lightweight adapter using the class token is enough to achieve comparable performance compared with the dual norm setting. It also enables model soup instead of model ensembling for faster model inference.

**Summary Of The Review:**

The idea is simple yet effective. However, it should be compared with the AdvProp in detail. It could only be applied on Transformers with class token, further limiting its application.

---

> ### Author Response · Authors · 2022-11-11
> **Thank you for the review**
>
> Thank you for the review. Please see answers below.
>
> > Even in the original AdvProp, the number of domain-specific parameters is marginally compared with the number of the whole model, which makes the benefit of further reducing the number of domain-specific parameters less useful.
>
> The main contribution of our work is not about reducing the number of domain-specific parameters (which is still nice to have) but (1) that we can use other layers than normalization layers as adapters and (2) that co-training with adapters allows model soups. This last point is not obvious at all as model soups were until now only used to ensemble several models fine-tuned from the same checkpoint.
>
>
> > model soup could also be applied to the AdvProp.
>
> We want to remind the reviewer that [Xi et al., 2020](https://arxiv.org/abs/1911.09665) does not study adversarial robustness and generalization under the lens of adapters. In fact, we argue that it wrongly claims that batch statistics are the key issue when doing co-training, which we demonstrate experimentally is not the case. That being said, we can indeed create model soups with other adapters (as shown in Table 1 and Figure 9). In the following, we apply model soups to the dual BatchNorm layers of the ResNet-50 model of Figure 2, which approximately corresponds to AdvProp except that we use untargeted attacks instead of the targeted attacks used in AdvProp. We report the clean and robust accuracy for the different soups in the table below. We observe a smooth transition between the clean and robust modes so model soups can effectively be applied to the dual BatchNorm layers of a co-trained ResNet model.
>
> | Beta | Clean Accuracy |  Robust Accuracy |
> | ----------- | ----------- | ----------- |
> |0   | 66.63% |  34.77% |
> |0.1 | 67.03% |  31.01% |
> |0.2 | 67.17% |  23.16% |
> |0.3 | 67.13% |  14.11% |
> |0.4 | 67.32% |  6.8  % |
> |0.5 | 68.18% |  2.36 % |
> |0.6 | 69.46% |  0.64 % |
> |0.7 | 70.99% |  0.17 % |
> |0.8 | 72.91% |  0.03 % |
> |0.9 | 75.98% |  0.01 % |
> |1   | 77.74% |  0    % |
>
> > [Using the class token as the adapter so it doesn't work with CNN and transformer without class token, which limits the application of this method.] [The influence of different adapters should be studied.]
>
> We study various adapters in both Table 1 and Figure 9 (in the appendix). In Table 1, we show that co-training with various adapters such as the embedder, the positional embedding, the LayerNorm parameters or the classification token results in similar performance both in clean and robust accuracy. Hence, we show that co-training works with other adapters than the classification token. Furthermore, we show in Figure 9 that adversarial model soups perform equally well for the various types of adapters. We have shown above that it also works for ResNets.
>
> > Comparison with AdvProp is not thorough. AdvProp could be regarded as using dual norm as the adapters.
>
> AdvProp can indeed be regarded as using dual normalization layers as adapters. Nevertheless, the AdvProp paper [Xi et al., 2020](https://arxiv.org/abs/1911.09665) focuses on splitting batch statistics and argues that differences in statistics are to blame for bad co-training behavior. As a result, recent work such as [Hermann et al., CVPR 2022](https://arxiv.org/abs/2111.15121) makes a different interpretation of AdvProp and states in the abstract that AdvProp is “not directly applicable to ViT”, supposedly due to the absence of batch statistics in the LayerNorm layers used in ViTs. In fact, [Xie et Yuille, 2019](https://arxiv.org/abs/1906.03787) from the AdvProp authors suggests similarly that GroupNorm is unaffected by co-training. Our work, which is in continuation with the AdvProp approach, corrects this interpretation by showing that separate batch statistics are not necessary and that domain-specific parameters are the key for good co-training performance. Hence, our work shed a new light on the mechanism of AdvProp and complement it by showing that (1) co-training works well with other layers as adapters and (2) that co-training with adapters enables model soups, which also apply to AdvProp.

---

> > ### Comment · Reviewer_VYKg · 2022-11-18
> > **Thanks for your response**
> >
> > Thanks to the authors for the response. It solves my question 1, 2, and 4, while the conditional token still has its limitation.
> > The model soups part has limited novelty for me, while the trainable parameters that could bridge the domain gap is interesting and novel.
> > I would raise the score to 6: marginal above the acceptance threshold.

---

### Official Review · Reviewer_rT7Z · 2022-10-28

**Confidence:** 4
**Correctness:** 3
**Technical Novelty And Significance:** 2
**Empirical Novelty And Significance:** 3
**Recommendation:** 6

**Clarity, Quality, Novelty And Reproducibility:**

This paper has fair clarity, quality, novelty and reproducibility. The main concerns about clarity and novelty are discussed in Weaknesses.

**Strength And Weaknesses:**

Strengths:

1. AdvProp (Xie et al. 2019a) improves image recognition by adversarial training with separate batch statistics of clean and adversarial data, first showing that adversarial examples can benefit model accuracy. This paper views AdvProp from a novel adapter perspective. It demonstrates that separating batch statistics is not necessary and that using domain-specific trainable parameters can achieve similar performance. The motivation is clear, and the new finding is interesting.

2. Compared to AdvProp that only consider clean performance, this paper takes both clean and robust performance into consideration. It introduces “adversarial model soups” that can trade off clean and robust accuracy via the adapter. This makes the models more flexible and applicable to more practical scenarios.

3. In addition to adversarial examples, this paper also considers the robustness against distribution shifts, where multiple ImageNet variant datasets are used for benchmarking, such as ImageNet-C and ImageNet-A. It is always good to evaluate on broad datasets.

4. Figure 2 is impressive. It first demonstrates the importance of the “dual” technique, then shows that AdvProp is not the only effective dual technique. Multiple normalization methods are evaluated. The results indicate that domain-specific trainable parameters are key, not batch statistics.


Weaknesses / Questions / Suggestions:

1. If my understanding is correct, the beta is a hyper-parameter of adversarial model soups that is manually adjusted. If that is true, it is less practical and less novel. At inference time, the given inputs would be any type (clean, adversarial, distribution shifts, etc). It is impossible to adjust the clean/adversarial modes or the beta value for each input sample. An automatic mechanism is needed. The model is expected to self-decide a proper mode or beta value for each input automatically with an end-to-end pipeline. In this case, the contribution of the “adversarial model soups” would be more significant. The author may refer to [r1], which combines an automatic selection module with separate batch normalization layers to achieve the idea. The author may figure out a way to automate the adversarial model soups. The current method is just a linear combination of domain-specific parameters, so the novelty is limited.

2. Several experimental results are strange. In Table 1, Co-training (Eq. 1) gets 0% robust accuracy, which is unexpected. With alpha=0.5, Co-training should still get robustness to a certain extent (see Xie & Yuille, 2019). Similarly, in Table 2, it is also unexpected that Fast-AT gets 0% robust accuracy. According to (Wong et al. 2020), Fast-AT can achieve decent robustness with proper step size and random initialization. The authors are asked to explain these results, otherwise, the results would be less convincing.

3. Several figures are not clear enough. Figure 1 is suggested to denote phi_clean, phi_adv, etc. (corresponds to Equations) on the figure to make it more understandable. Figure 4 (similarly, Figure 9 in the appendix) should provide the corresponding beta values like Figure 3 provides alpha values. Figure 5 should self-contain the x-axis tile (beta).

4. The tables of experimental results can be more complete. Specifically, the section “Robustness to stronger attack” should have a table showing the numbers, which would be much more clear. Or expand columns in Table 1 for stronger attacks. Furthermore, Table 9 (in the appendix) should compare the numbers of adversarial model soups as well.

[r1] S.-Y. Lo and V. M. Patel, “Defending Against Multiple and Unforeseen Adversarial Videos,” in IEEE Transactions on Image Processing, 2021.

**Summary Of The Paper:**

This paper introduces a new finding that is contrary to previous works. That is, it is not necessary to separate batch statistics when co-training on clean and adversarial data. It shows that using the classification token of a Vision Transformer (ViT) as an adapter is sufficient to achieve the performance of the dual normalization layers proposed by previous works. This paper also introduces “adversarial model soups” that allow a smooth transition between the clean and robust modes. These observations provide new insights into the regularization effect of adversarial training.

**Summary Of The Review:**

Based on the above comments, I suggest marginally below the acceptance threshold. The new finding is interesting, and the experiments are extensive. However, the novelty of adversarial model soups is not significant. Furthermore, there are some unexpected experimental results that need further explanation. My final rating will depend on the authors’ response.

---

> ### Author Response · Authors · 2022-11-11
> **Thank you for the review**
>
> Thank you for the review. Please see answers below.
>
> > [It is impossible to adjust the clean/adversarial modes or the beta value for each input sample] [An automatic mechanism is needed]
>
> In our study,  we have shown that it is possible to find a soup that works well across distribution shifts (see Figure 5). In many practical applications, there exists a small validation set (representative of the task) that could allow practitioners to find the best model soup. As mentioned, we could also adapt the soup during deployment or use a mixture of soups that automatically selects the right soup for each sample. However, integrating it in our pipeline would be another paper in itself and we remind the reviewer that we are limited to nine pages. We have added this idea to the paragraph about future work.
>
> > [experimental results are strange] [Co-training should still get robustness to a certain extent (see Xie & Yuille, 2019)]
>
> [Xie et Yuille, 2019](https://arxiv.org/abs/1906.03787) use random targeted attacks whereas we use stronger untargeted attack, which is the de facto standard attack type used in state-of-the-art $\\ell_p$-norm robustness literature (according to [RobustBench](https://robustbench.github.io/)). To be more comparable with random targeted attacks, we have run co-training without adapters for untargeted attacks with smaller perturbation radii: $\\epsilon_\\infty \in \\{0.5/255, 1/255, 2/255\\}$. We observe for these perturbation radii that co-training can still get some robustness at the end of training. We also note that $\\epsilon_\\infty \in \\{1/255, 2/255\\}$ still suffer from a strong decrease in robust accuracy at some point during training, illustrating the instability of co-training without adapters. We have added these experiments to the appendix in Figure 10.
>
> > it is also unexpected that Fast-AT gets 0% robust accuracy.
>
> We took a step size of 1.25$\\epsilon$ for Fast-AT as advised in their paper. Based on the reviewer’s suggestion we have tried smaller step sizes of $\\epsilon$ and 0.75$\\epsilon$, leading to improved results for Fast-AT which now matches PGD$^2$ in clean accuracy and is -1.38\% worse in robust accuracy. This observation corroborates the finding made in the Fast-AT paper by [Wong et al., 2020](https://arxiv.org/pdf/2001.03994.pdf) that large step sizes can result in catastrophic overfitting. Similarly, we have re-evaluated N-FGSM with smaller step sizes than advised in the corresponding paper and this also leads to improved results (but still worse than PGD$^2$ for robust accuracy). We have modified our manuscript to take these new results into account and we have highlighted that smaller step sizes are required when co-training with one-step attacks on ImageNet. Overall, these results still confirm our choice of PGD$^2$.
>
> > Several figures are not clear enough
>
> We thank the reviewer for these suggestions and we have modified Figure 1, 4, 5 and 9 accordingly.
>
> > [the section “Robustness to stronger attack” should have a table] [Table 9 ‘...’ should compare the numbers of adversarial model soups]
>
> In the revised paper we have added Table 10 which contains the robustness numbers against AA+MT for the various co-training loss weightings which correspond to the Pareto front when compromising between clean and robust accuracy. We have completed Table 9 with the clean and robust accuracies for adversarial model soups as suggested.
>
> > novelty of adversarial model soups is not significant.
>
> We would like to highlight that our proposed adversarial model soups significantly differ from the original model soups proposed by [Wortsman et al., 2022](https://arxiv.org/abs/2203.05482). First, whereas the original model soups are used to ensemble several models fine-tuned from the same checkpoint, our work shows that model soup is possible for a network co-trained with adapters. Thus, we show that model soups can be used beyond their original use case of ensembling fine-tuned models. Second, the soups of adapters merge the nominal and robust behaviors of the robust and clean mode respectively, while [Wortsman et al al. 2022] only consider nominal models. Both these innovations could inspire new research directions beyond the adversarial training setting such as co-training on multiple domains or datasets for multi-task learning.

---

> > ### Comment · Reviewer_rT7Z · 2022-11-21
> > **Thanks for the responses**
> >
> > Thanks for the responses. I'm satisfied with most of the responses. In particular, Tables 2, 9, 10, and Figures 1, 4, 5 and 9 have been revised accordingly. However, the novelty still prevents me from giving a very high score. Overall, I decided to raise the score from "5: marginally below the acceptance threshold" to "6: marginal above the acceptance threshold".

---

### Decision · Program_Chairs · 2023-01-20

**Decision:**

Accept: notable-top-25%

**Justification For Why Not Higher Score:**

Firstly, as stated in the next question, the quality of this paper is definitely above an average ICLR poster.

The main reason to select it as a "spotlight" rather than an "oral" is that its technical novelty part is not particularly strong, and it is unclear (though could still be promising) to generalize this method to models without class tokens.

**Justification For Why Not Lower Score:**

The quality of this paper is definitely above an average ICLR poster: the proposed method is interesting, and the empirical results are strong; in addition, I agree with the reviewers that this paper could inspire more future works, as it provides new insights into the regularization effect of adversarial training.

**Metareview: Summary, Strengths And Weaknesses:**

This paper continues the exploration of using adversarial examples to improve recognition models. Unlike previous methods, which heavily rely on separate batch statistics to cope with the joint training of adversarial examples and clean images, this paper demonstrates that the classification token in ViT can be used for this purpose and is more parameter-efficient. Additionally, this approach allows for easy integration with "model soups" to achieve the desired trade-off between clean and robust accuracy.

Overall, the reviewers enjoyed reading this paper, and found this method simple and effective. The reviewers only had some minor concerns about empirical evaluations. These concerns are well addressed in the rebuttal. As a result, all reviewers unanimously agree to accept this submission.


**Note From Pc:**

if the above contains the word "oral" or "spotlight" please see: "oral" presentation means -> notable-top-5% and "spotlight" means -> notable-top-25%. As stated in our emails, we are disassociating presentation type from AC recommendations